# Impact of interhospital competition on mortality of patients operated on for colorectal cancer faced to hospital volume and rurality: A cross-sectional study

Seydou Goro[1,2,3], Alexandre Challine[1,2,4,5]*, Jérémie H. Lefèvre[4,5], Salomé Epaud[6], Andrea Lazzati[3,7]

**1** Université Paris Cité, Paris, France, **2** HeKA, Inria, Paris, France, **3** Service de chirurgie digestive, Centre Hospitalier Intercommunal de Créteil, Créteil, France, **4** Service de chirurgie digestive, AP-HP, Hôpital Saint Antoine, Paris, France, **5** Sorbonne Université, Paris, France, **6** KADUCEO, Toulouse, France, **7** Université Paris Est Créteil, Créteil, France

* alexandre.challine@aphp.fr

## Abstract

### Introduction

Contradictions remain on the impact of interhospital competition on the quality of care, mainly the mortality. The aim of the study is to evaluate the impact of interhospital competition on postoperative mortality after surgery for colorectal cancer in France.

### Methods

We conducted a retrospective cross-sectional study from 2015 to 2019. Data were collected from a National Health Database. Patients operated on for colorectal cancer in a hospital in mainland France were included. Competition was measured using number of competitors by distance-based approach. A mixed-effect model was carried out to test the link between competition and mortality.

### Results

Ninety-five percent (n = 152,235) of the 160,909 people operated on for colorectal cancer were included in our study. The mean age of patients was 70.4 ±12.2 years old, and female were more represented (55%). A total of 726 hospitals met the criteria for inclusion in our study. Mortality at 30 days was 3.6% and we found that the mortality decreases with increasing of the hospital activity. Using the number of competitors per distance method, our study showed that a "highly competitive" and "moderately competitive" markets decreased mortality by 31% [OR: 0.69 (0.59, 0.80); p<0.001] and by 12% respectively [OR: 0.88 (0.79, 0.99); p<0.03], compared to the "non-competitive" market. High hospital volume (100> per year) was also associated to lower mortality rate [OR: 0.74 (0.63, 0.86); p<0.001].

**Data Availability Statement:** A regards data availability, due to the sensitive nature of the data that support the findings, access to them is

restricted and can only be granted by the Ethics and scientific committee for health research, studies, and evaluations (CESREES, Comite Ethique et Scientifique pour les Recherches, les Etudes et les Evaluations dans le domaine de la Sante) and/or the French data protection authority (Comite National de l'Informatique et des Libertes, CNIL), and so are not readily available. The data are part of the National health data system (SNDS, Système national des Donnees de Sante) a database maintained by the HDH (Health Data Hub). The HDH can be contacted. https://www.health-data-hub.fr/contact.

**Funding:** The author(s) received no specific funding for this work.

**Competing interests:** The authors have declared that no competing interests exist.

## Conclusions

The results of our studies show that increasing hospital competition independently decreases the 30-day mortality rate after colorectal cancer surgery. Hospital caseload, patients' characteristics and age also impact the post-operative mortality.

## Introduction

The interest around competition in healthcare is growing in public health analysis, as according to economic theories, competition would increase the quality of care when prices in the market are regulated [1, 2].

Several countries, including France, introduced policies in recent years to encourage competition between hospitals [3–7]. In France, the hospital healthcare system is publicly funded. There are three types of hospital ownership: public, private not-for-profit, and private for-profit hospitals. Only the first type is entirely state-owned. From 2004 to 2008, pro-competition reforms were gradually introduced [8]. Prior to this reform, public and not-for-profit hospitals were paid from a global budget, while private for-profit institutions were reimbursed on a fee-for-service basis.

The reform led to the introduction of a payment system based on Diagnostic Related Groups (DRGs) for all types of hospitals, including private for-profit facilities. The DRG-system consists of a defined amount of reimbursement depending on the diagnoses, procedures, patient's comorbidities and complications [4]. In this context of regulated prices, hospitals will be encouraged to improve the quality of care in order to attract more patients. As measuring the quality of care is complex and multidimensional, in-hospital mortality is often used as a proxy of hospital quality of care [9].

However, the results of empirical studies remain contradictory. Some studies found a positive impact of competition on quality of care when prices are set [10–12] while other studies report a negative impact [13, 14]; and others found contradictory effects depending on the quality of care or on the competition indicators that were used [15, 16]. Finally, some authors did not find any relationship between competition and quality of care [17].

In France, very few studies have been carried out on the impact of competition on in-hospital mortality. A study by Gobillon et al. in 2017 on patients admitted for an acute myocardial infarction (AMI) showed a beneficial impact of competition on mortality in not-for-profit hospitals [4]. Nevertheless, generalizability of these results to other disciplines may be limited, as several studies have reported that the association between competition and quality of care is sensitive to the pathology concerned [15, 16].

To date, no studies have been carried out on the impact of inter-hospital competition on mortality after digestive surgery, particularly for colorectal cancer, which represents a major public health issue, as in 2018, more than 43,336 new cases were diagnosed, and more than 17,000 people died in France, representing the fourth leading cause of cancer and the second leading cause of cancer mortality [18].

The aim of this study was to assess the specific impact of inter-hospital competition on post-operative mortality in patients operated for colorectal cancer considering hospital volume and rurality.

## Methods

### Study design

Data were extracted from a National Health Database (the "*Programme de Médicalisation des Systèmes d'Informations*", PMSI), which is a mandatory billing system for all hospitalization in France, either for public or private hospital [19]. The aim of the PMSI is to set the budget of hospitals based on hospital activity. The external validity of this database has been tested with other national registry [20–22]. Access to the database was requested from and granted by the National Commission on Informatics and Liberty (*"Commission Nationale de l'Informatique et des Libertés"*, CNIL). As the identifier is anonymous, patient consent was not required. This study complied with MR005 (*'''*Méthodologies de Référence") and the access to the database was submitted for authorization by the National Commission on Informatics and Liberty (CNIL, authorization n° 01947391).

Our study consisted of a cross-sectional analysis of the data over the period from 2015 to 2019. We included in the study all patients operated on for either colon cancer, cancer of the rectosigmoid junction or rectal cancer, (ICD-10 codes: C18-C20) living in mainland France, without any age restriction. Patients residing in the French overseas department and all patients treated in a hospital located in these regions were excluded. Overseas territories do not have the same health and competition issues as the rest of France because of their geographical specificity. Patients from foreign countries were also excluded. Hospitals with less than five patients with colorectal cancer treated over the 5 years (i.e., one patient per year on average) were excluded, as it is difficult to determine the markets by patient flow for hospitals with extremely low recruitment.

### Measures of competition and definition of markets

Competition is a rivalry where two or more parties strive for a common goal. The competition faced by a hospital is usually measured either by the number of competitors in its market or by the Herfindalh-Hirshman index (HHI). The HHI is calculated by summing the square of the market share of each hospital recruiting in that area. For example, in case of four different hospitals recruiting in the same area, with a market share of 40%, 30%, 20% and 10% respectively, the HHI will be: $(0,4)^2 + (0,3)^2 + (0,2)^2 + (0,1)^2 = 0,3$; [23]. The lower the HHI, the higher competition is. The key issue before measuring the competition index for each hospital is to define is the limits of hospitals market. Some authors have proposed the method based on a fixed radius around the hospital, while others have proposed the method based on patient flow, i.e. a radius containing X% of the hospital recruitment. This last method thus makes it possible to define variable radius markets according to the hospital's activities.

For the main analysis of our study, we measure competition using the method of the number of competitors in fixed radius around the hospital. Firstly, for each hospital we define the recruitment area, based on a radius of 30 km around the given hospital. This distance was previously used by Gobillon *et al*, in their study on AMI in France [4]. Then, all other hospitals included in this area and recruiting at least 1% of their total admission in that area, were considered as competitors. The 1% threshold is arbitrary and has been used to exclude from competition all hospitals present in the market but with insignificant activities. Hence, the measure of competition for each hospital is defined by the number of competitors in its recruitment area. The competition was categorized in four levels using the interquartile intervals as "non-competitive", "slightly competitive", "moderately competitive" and "highly competitive".

This method of assessing competition has been criticized because a hospital located outside a given recruitment area and recruiting a significant number of patients in that area would not

be considered as a competitor [23]. For this reason, in sensitivity analysis, we employed the patient flow method to determine the hospital market. For each hospital, we identified all municipalities providing at least 3% of the total admissions or participating to 40% of cumulative admissions. Thus, all municipalities meeting these criteria constitute the hospital market. These criteria were previously reported by Palangkaraya et al. [14]. Municipalities providing only one patient were excluded, in order to minimize the bias of occasional recruitment (as for example the case of a holidaymaker treated in another place far from his place of residence).

### Study outcome and covariates

The primary outcome of the study was the 30-day mortality rate after the surgery. We controlled for patients' clinical data (surgical procedure, Charlson comorbidity index, emergency admission, and patient history of malnutrition, obesity and neo-adjuvant treatment), and demographic data (age, sex, French Deprivation Index, version of 2009) and for hospital's characteristics (Hospital caseload, and status). Charlson comorbidity index (the version of Quan and colleagues) was calculated based on the patient's overall diagnosis, using the International Classification of Diseases, 10th version (ICD-10), in the last two years prior to surgery (S1 Appendix) [24, 25]. The surgical procedures were defined using the procedures classification, CCAM, including all surgical, endoscopic and radiological interventions (S2 Appendix). The history of obesity, neo-adjuvant treatment and malnutrition were defined using also ICD-10 codes (S3 Appendix). Coding algorithms are available in S1–S3 Appendices. The French Deprivation Index (FDep) is a geographical indicator that measures precariousness within the patient's place of residence. It is composed of four components: *(1)* Unemployment rate in the labor force (i.e., 15–64 years old in France), *(2)* rate of workmen in the labor force, *(3)* rate of high school graduates in the out-of-school population over 15 years of age and *(4)* reported median income per consumption unit [26]. It is therefore an indirect measure of the patient's level of deprivation. Hospitals were classified as teaching hospital (University hospital) and non-teaching hospital composed of public, non-profit and for-profit hospital. The rural or urban nature of the hospital area was thus approximated by the density of the area from which the patients attending the hospital came. This variable was categorized by quartile in four levels of density.

### Statistical analysis

We described all numeric variables with the mean and standard error and the categorical variable with numbers and percentages. Assuming that patients treated in the same hospital may be more correlated with each other than those treated in different hospitals, we fitted a logistic mixed-effect model that considers this correlation, and we adjust for hospital random effect. In the main analysis, we studied the relationship between the number of competitors (categorized) and 30-day mortality. In order to verify the robustness of our model, we performed two sensitivity analyses using the HHI (by patient-flow method) as a measure of competition. All p-values presented were for a 2-sided test, and the threshold of significance was set at $P < 0.05$. All analyses were performed with the R statistical software package (R Foundation for Statistical Computing, Vienna). Maps were drawn with *Leaflet* R-package (https://CRAN.R-project.org/package=leaflet).

### Results

During the study period (2015–2019), a total of 160,909 patients were operated for colorectal cancer in France. Among these, 8,674 (5.4%) patients did not meet inclusion criteria and were excluded (Fig 1). Six hundred and seventy-six (0.4%) of eligible patients were removed for

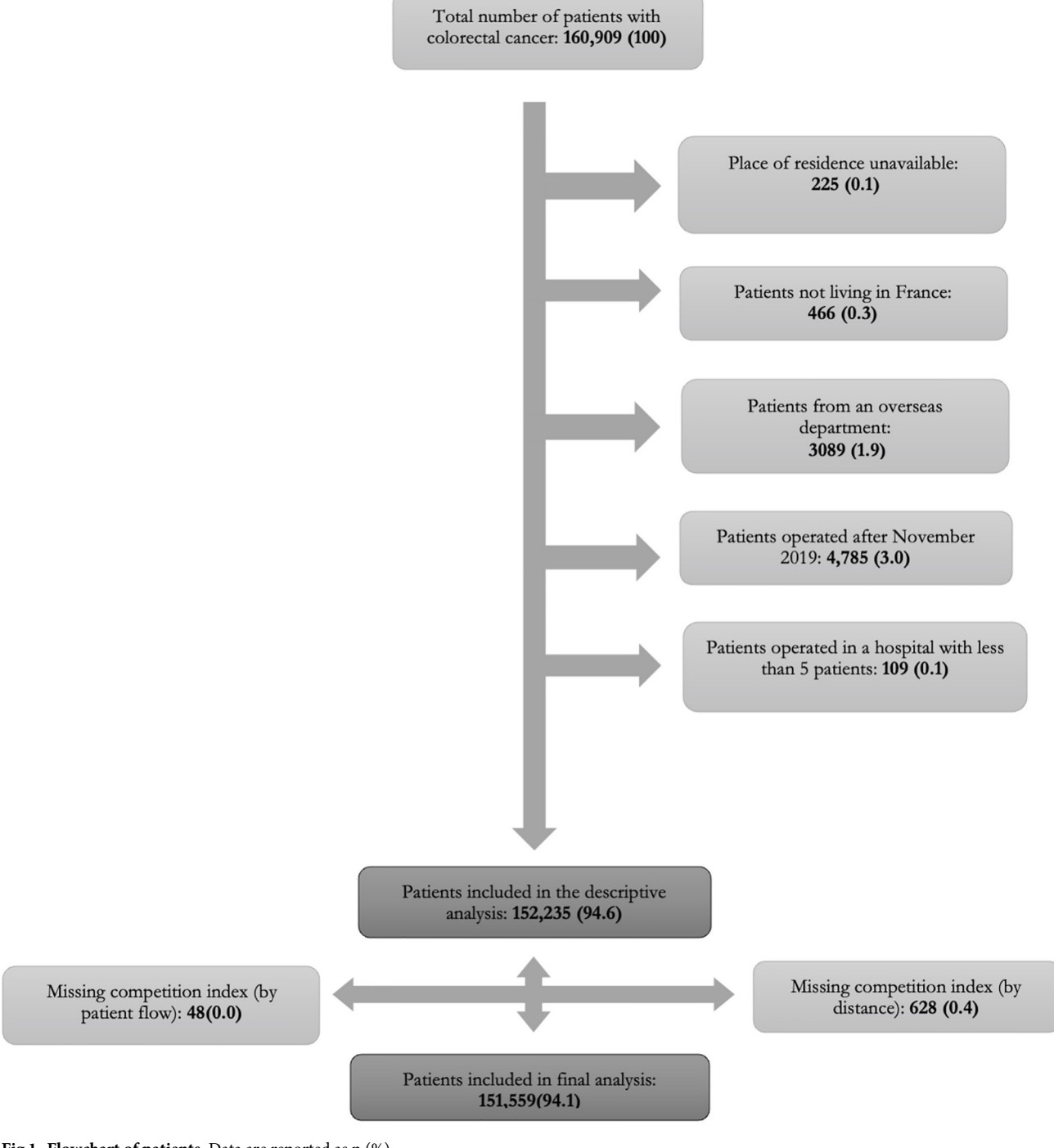

**Fig 1. Flowchart of patients.** Data are reported as n (%).

missing data. The final number of patients included for analysis was 151,559. The mean age of patients was 70.4 ±12.2 years old, and female were more represented (55%). Patients' characteristics are detailed in Table 1.

A total of 726 hospitals met the criteria for inclusion in our study, of which 8% were public teaching hospitals. Most of the hospitals (68.9%) operated less than 50 patients/year over the 5

**Table 1. Baseline characteristics of patients.**

| Covariates | | Total n = 152,235 |
|---|---|---|
| **Gender** | Female | 67,647 (44.4) |
| **Age, mean (SD)** | | 70.4 (12.2) |
| **Charlson index** | 0–2 | 114,220 (75.0) |
| | 3–4 | 19,839 (13.0) |
| | >4 | 18,176 (11.9) |
| **Obesity** | | 18,919 (12.4) |
| **Malnutrition** | | 32,981 (21.7) |
| **Surgical procedure** | Right colectomy | 51,269 (33.7) |
| | Left colectomy | 35,234 (23.1) |
| | Rectosigmoid resection | 34,436 (22.6) |
| | Rectal resection | 10,798 (7.1) |
| | Multiple resection | 13,221 (8.7) |
| | Other | 7,277 (4.7) |
| **Emergency admission** | | 15,752 (10.3) |
| **Neo-adjuvant treatment** | | 18,696 (12.3) |
| **LOS, mean (SD)** | | 13.5 (11.3) |
| **FDep09** | Lower category | 45,212 (29.7) |
| | Middle category | 48,703 (32.0) |
| | Upper category | 58,320 (38.3) |

Data are reported as n (%) unless otherwise stated. LOS: length of stay, FDep: French Deprivation Index

years (Table 2). Competition indicators using the radius method could not be calculated for 8 hospitals (1.1%) because these hospitals had no recruitment within a distance of 30 km. Competition indicators using the patient flow method could not be calculated for seven hospitals (1%) because the low recruitment per municipality.

The overall mortality rate at 30 days after surgery was 3.6%. Using the number of competitors method, mortality in non-competitive, slightly competitive, moderately competitive and highly competitive market was respectively 4.4%, 3.9%, 3.3% and 2.4%, p<0.001 (Fig 2). With the HHI method mortality was identical for all levels except for highly competitive markets where it showed a significant decrease (from 3.9% to 2.9%, p<0.001) (Fig 3).

In multivariate analysis, factors significantly associated to mortality were age, gender, patient comorbidity index, malnutrition, and neo-adjuvant treatment, the emergency admission and type of surgical procedures. Hospital caseload was also associated with the mortality (Table 3).

After controlling for covariates, we observed a significant lower mortality in the moderately competitive market and in highly competitive market compared to the non-competitive market (OR: 0.88, 95%CI 0.78–0.99, p = 0.03; OR: 0.69, 95%CI 0.59–0.80, p<0.001 respectively).

We had similar result with HHI by patient flow method. The association between competition and mortality was significant only for the highly competitive markets compared to the non-competitive markets [OR: 0.81 (0.72–0.92), p<0.001] (Table 3).

The volume of activity was also associated with the 30-day-mortality rate. The mortality was significantly lower in high volume or mid volume hospital compared to low volume hospital (OR: 0.74, 95%CI 0.63–0.86, p<0.001; OR: 0.87, 95%CI 0.80–0.96, p = 0.004 respectively) (Table 3).

## Discussions

In this study, based on public data, we assessed the impact of inter-hospital competition on 30-day mortality after colorectal cancer resection, on more than 150,000 patients on a

**Table 2. Characteristics of hospitals and those of their markets.**

| Covariates | N° of hospitals n = 726 | N° of patients n = 152 235 |
|---|---|---|
| **Hospital's status** | | |
| Teaching hospital | 61 (8.4) | 24,896 (16.4) |
| Non-teaching hospital | 665 (91.6) | 127,339 (83.6) |
| **Hospital caseload/years** | | |
| $\leq 50$ | 500 (68.9) | 58,552 (38.5) |
| 51–100 | 179 (24.6) | 62,696 (41.2) |
| > 100 | 47 (6.5) | 30,987 (20.3) |
| **Number of competitors** | | |
| Non-competitive (0–1) | 298 (41.1) | 45,774 (30.1) |
| Slightly competitive (2–3) | 178 (24.5) | 38,982 (25.6) |
| Moderately competitive (4–8) | 144 (19.8) | 35,460 (23.3) |
| Highly competitive (9–36) | 98 (13.5) | 31,391 (20.6) |
| NA | 8 (1.1) | 628 (0.4) |
| **HHI** | | |
| Non-competitive (0.3085–0.5535) | 179 (24.6) | 44,611 (29.3) |
| Slightly competitive (0.2288–0.3085) | 183 (25.2) | 38,133 (25.0) |
| Moderately competitive (0.1535–0.2288) | 179 (24.6) | 28,017 (18.4) |
| Highly competitive (0.0268–0.1535) | 178 (24.5) | 41,426 (27.2) |
| NA | 7 (1.0) | 48 (0.03) |
| **Density of hospital area** | | 37888 |
| Very rural (2.29–83.7) | | 37903 (25%) |
| Rural (83.7–264) | | 37955 (25%) |
| Urban (264–1660) | | 37807 (25%) |
| Very urban (> 1660) | | 37888 (25%) |

Data are reported as n (%) unless otherwise stated. HHI: Herfindahl-Hirschman index. Density of patient (inhabitant/km$^{22}$)

nationwide scale. The overall post-operative mortality was 3.6% and it was associated to competition: the more a hospital was confronted with competition, the less mortality was observed regardless of the activity volume and rurality of hospital. This relation is significant for hospital evolving in highly competitive zones. We also found a significant effect of hospital caseload, patients' characteristics and age on mortality. These findings are consistent with previous studies, El Amrani and al. found an association between lower 90-day post-operative mortality and higher volume activity [27–29].

Concerning the impact of competition, other studies found a beneficial impact on the mortality using different methods. This is the case for the study carried out by Gaynor and al., also using patient flow method, on patient admitted for AMI in United Kingdom. According to this study a 10% increase in the HHI leads to an increase of 2.46% in the AMI death rate [30, 31]. Similarly, the study conducted by K.F. Erickson et al. on public data from 2001 to 2009 in the USA shows that decrease market competition led to an increase of mortality for hemodialysis patients [10]. Although the study by Gobillon et al., carried out in patients with AMI in France, does not show a link between inter-hospital competition and mortality in general, it should be noted that it did, however, highlight the beneficial effect of competition on mortality for not-for-profit hospitals [4]. Another aspect of Gobillon's study is that it compares the impact of competition on mortality between two periods, the period before the pro-competitive reform and the period after this reform [4].

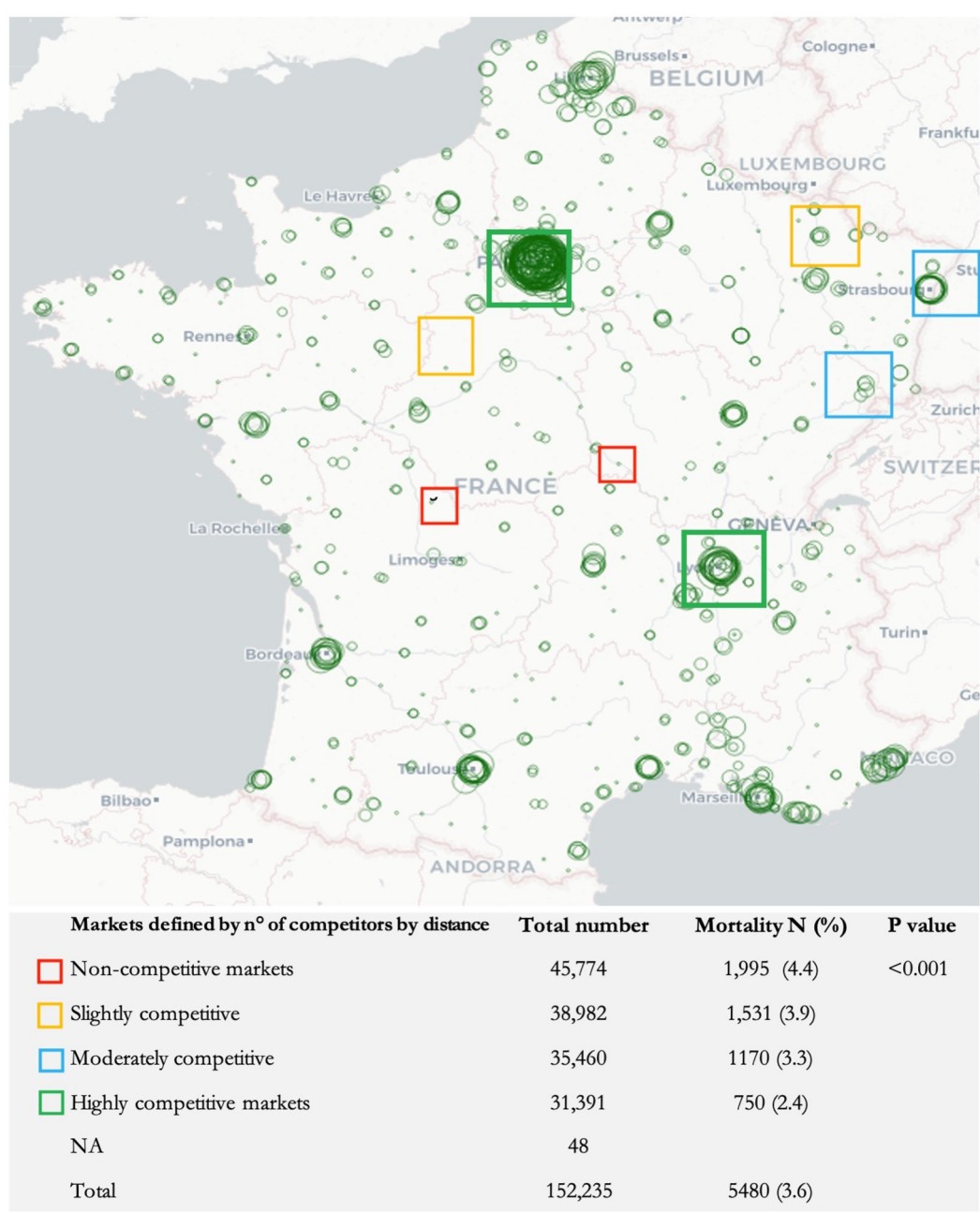

| Markets defined by n° of competitors by distance | Total number | Mortality N (%) | P value |
|---|---|---|---|
| Non-competitive markets | 45,774 | 1,995 (4.4) | <0.001 |
| Slightly competitive | 38,982 | 1,531 (3.9) | |
| Moderately competitive | 35,460 | 1170 (3.3) | |
| Highly competitive markets | 31,391 | 750 (2.4) | |
| NA | 48 | | |
| Total | 152,235 | 5480 (3.6) | |

**Fig 2. Representation of market types (by distance method) and their mortalities.** This figure illustrates the geographical distribution of hospitals and the number of hospitals competing in their market (i.e., the intensity of competition). For example, most of the hospitals in the green rectangle have high competition in their market. The table below shows the average mortality in each type of market. Leaflet | ©.

However, other studies show that competition increases mortality [13, 14]. The study conducted by Joel T. Adler et al, on the impact of inter-hospital competition on patients receiving a kidney transplant, shows that increased competition increases mortality and graft failure, when the donor is a deceased person. This correlation in their study might be explained by a mediating factor, which is the use of lower quality grafts, since they also found a positive

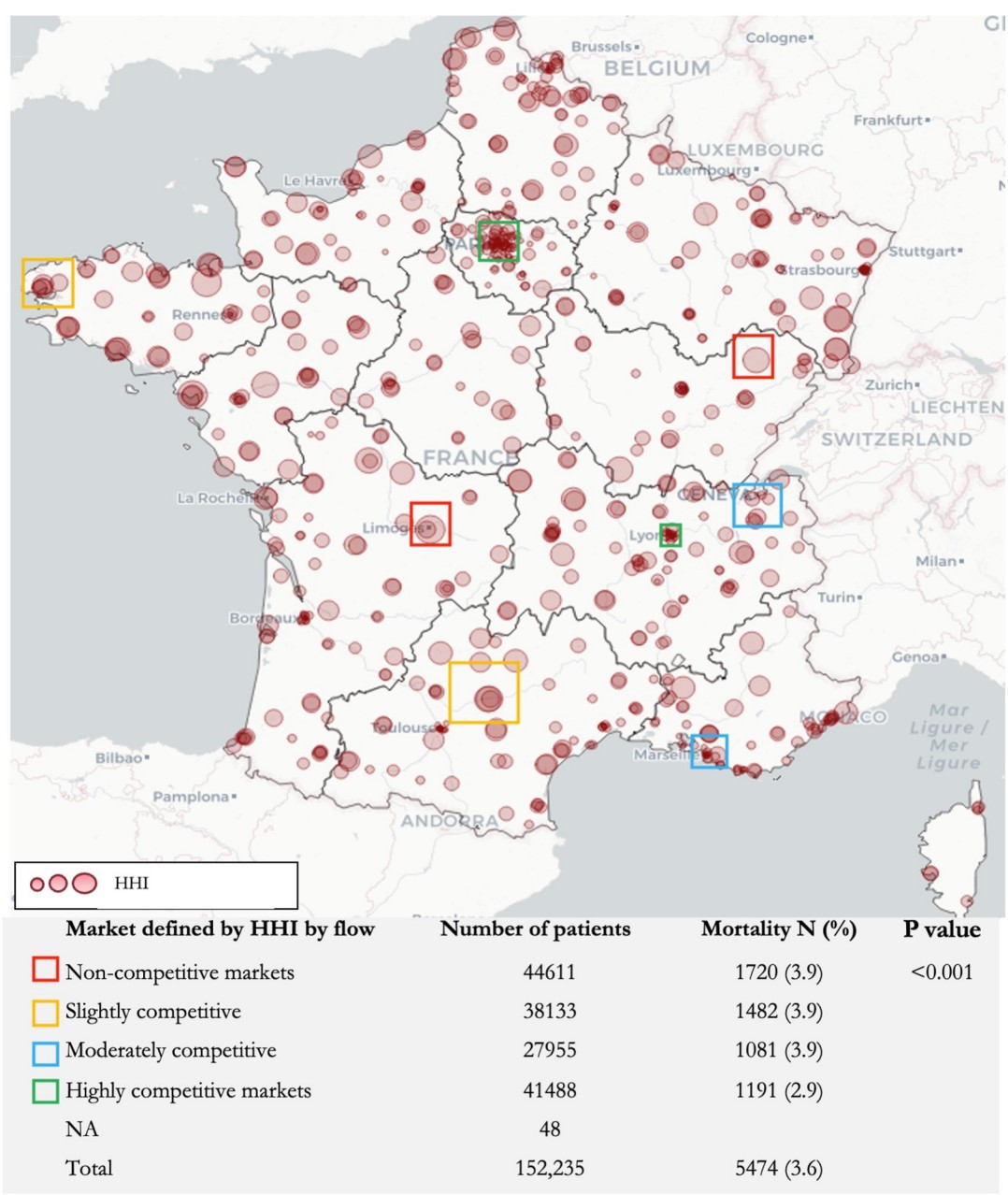

| Market defined by HHI by flow | Number of patients | Mortality N (%) | P value |
|---|---|---|---|
| Non-competitive markets | 44611 | 1720 (3.9) | <0.001 |
| Slightly competitive | 38133 | 1482 (3.9) | |
| Moderately competitive | 27955 | 1081 (3.9) | |
| Highly competitive markets | 41488 | 1191 (2.9) | |
| NA | 48 | | |
| Total | 152,235 | 5474 (3.6) | |

**Fig 3. Representation of market types (by patient-flow method) and their mortalities.** This figure shows the geographic distribution of hospitals and the Herfindalh-Hirshman Index in their market. The smaller the HHI, the greater the competition. For example, most of the hospitals in the orange rectangles have a slightly competitive market while those in the red rectangles have a non-competitive market. The table below shows the average mortality in each type of market. Leaflet | ©.

correlation between competition and the use of poor-quality kidneys. This discrepancy with our findings could also be explained by the absence of hospital volume in the multivariable Cox model of this previous study [13]. Indeed, the activity volume in a non-competitive hospital may be higher than in a highly competitive center. As surgical activity is a major factor associated to the post-operative mortality, it is very important to take this into account [27–29].

**Table 3. Odds ratios of relationship between competition and 30-day mortality using patient flow methods.**

| Covariates | Number of competitors by distance method | | HHI by patient flow method | |
|---|---|---|---|---|
| | OR [CI] | p.value | OR [CI] | p.value |
| **Market competitiveness** | | | | |
| Non-competitive | —— | —— | —— | —— |
| Slightly competitive | 1.02 (0.92, 1.13) | 0.71 | 0.98 (0.88, 1.09) | 0.70 |
| Moderately competitive | 0.88 (0.78, 0.99) | 0.03 | 0.97 (0.86, 1.09 | 0.60 |
| Highly competitive | 0.69 (0.59, 0.80) | <0.001 | 0.80 (0.70, 0.91) | <0.001 |
| **Age (years)** | —— | —— | —— | —— |
| ≤60 | | | | |
| 61–70 | 1.37 (1.22, 1.53) | <0.001 | 1.37 (1.23, 1.53) | <0.001 |
| 71–80 | 1.86 (1.68, 2.08) | <0.001 | 1.87 (1.68, 2.08) | <0.001 |
| 81–90 | 3.63 (3.27, 4.03) | <0.001 | 3.64 (3.28, 4.04) | <0.001 |
| > 90 | 6.42 (5.59, 7.37) | <0.001 | 1.37 (1.23, 1.53) | <0.001 |
| **Gender** | —— | —— | —— | —— |
| Female | | | | |
| Male | 1.36 (1.29, 1.44) | <0.001 | 1.36 (1.29, 1.44) | <0.001 |
| **Charlson comorbidity index** | | | | |
| 0–2 | —— | —— | —— | —— |
| 3–4 | 1.30 (1.20, 1.41) | <0.001 | 1.30 (1.20, 1.41) | <0.001 |
| >4 | 2.43 (2.25, 2.62) | <0.001 | 2.42 (2.24, 2.61) | <0.001 |
| **Denutrition** | | | | |
| No | —— | —— | —— | —— |
| Yes | 1.42 (1.34, 1.51) | <0.001 | 1.42 (1.34, 1.51) | <0.001 |
| **Emergency admission** | | | | |
| No | —— | —— | —— | —— |
| Yes | 2.93 (2.73, 3.14) | <0.001 | 2.94 (2.74, 3.15) | <0.001 |
| **Neo-adjuvant treatment** | | | | |
| No | —— | —— | —— | —— |
| Yes | 1.23 (1.11, 1.35) | <0.001 | 1.23 (1.11, 1.35) | <0.001 |
| **Surgical procedure** | | | | |
| Right colectomy | —— | —— | —— | —— |
| Left colectomy | 0.86 (0.80, 0.93) | <0.001 | 0.86 (0.80, 0.93) | <0.001 |
| Rectal resection | 0.67 (0.58, 0.79) | <0.001 | 0.67 (0.58, 0.78) | <0.001 |
| Multiple resection | 1.08 (0.98, 1.20) | 0.1229 | 1.08 (0.98, 1.20) | 0.12 |
| Recto-sigmoid resection | 0.84 (0.77, 0.92) | <0.001 | 0.84 (0.77, 0.92) | <0.001 |
| Other procedures | 1.55 (1.39, 1.72) | <0.001 | 1.55 (1.39, 1.72) | <0.001 |
| **Hospital's caseload/year** | | | | |
| < = 50 | —— | —— | —— | —— |
| 51–100 | 0.87 (0.80, 0.96) | 0.0040 | 0.84 (0.76, 0.92) | <0.001 |
| >100 | 0.74 (0.63, 0.86) | <0.001 | 0.69 (0.59, 0.80) | <0.001 |
| **Hospital's status** | | | | |
| Teaching hospital | —— | —— | —— | —— |
| Non-teaching hospital | 0.90 (0.78, 1.03) | 0.12 | 0.90 (0.77, 1.04) | 0.14 |
| **French Deprivation index** | | | | |
| Lower category | —— | —— | —— | —— |
| Middle category | 1.00 (0.93, 1.08) | 0.98 | 1.00 (0.93, 1.07) | 0.98 |
| Upper category | 0.98 (0.90, 1.06) | 0.54 | 0.95 (0.88, 1.03) | 0.24 |
| **Density of hospital area** | | | | |

*(Continued)*

**Table 3.** (Continued)

| Covariates | Number of competitors by distance method | | HHI by patient flow method | |
|---|---|---|---|---|
| | OR [CI] | p.value | OR [CI] | p.value |
| Very rural | —— | —— | —— | —— |
| Rural | 1.03 (0.95, 1.11) | 0.52 | 1.03 (0.95, 1.12) | 0.42 |
| Urban | 1.01 (0.92, 1.09) | 0.89 | 1.00 (0.92, 1.09) | 0.92 |
| Very urban | 1.09 (0.99, 1.21) | 0.07 | 1.06 (0.96, 1.16) | 0.26 |

Data are reported as n (%) unless otherwise stated. HHI: Herfindahl-Hirschman index. Density of hospital area (inhabitant/km$^{2}$)

We used different methods to determine hospital market and to measure competition in this market. In our main analysis we determine hospital market by distance method and then measure the competition in this market by the number of competitors. We also used HHI by patient flow method to check the robustness of our method. However, the methods most often used in the literature for market assessment are those based on geopolitical boundaries (Donation Service Area, Health Service Area, or county. . .) and fixed radii around the hospital (or distance-based method) because of their ease of implementation [4, 10, 13, 15, 23]. But these methods have certain limitations. In a geopolitically defined market, a neighboring hospital is not considered a competitor when it is on the other side of the border, while two hospitals that are far apart will be considered competitors when they are in the same geopolitical area. The distance-based method overcomes this limitation by including competitors that are close to a hospital but located on the other side of a geopolitical border. Nevertheless, in the case of a fixed radius approach (the method we used) the same catchment area is assigned to all hospitals whether they are in urban area or a rural area, while the rural hospitals naturally have a larger catchment area than the urban hospitals. Patient flow approaches, which are variable radius methods, have the advantage of largely overcoming the limitations of the first two methods. The number of competitors and the Herfindahl Hirschman Index are also the most widely used indicators of competition. The latter has the advantage of taking into account the weight of hospitals in the sharing of patients in the market compared to the number of competitors, which assigns the same importance to all hospitals whether they have intense or minimal activity in the market [23].

Despite the robustness of the results of our study, it should be noted that it is subject to certain limitations. We were not able to calculate competition indicators for a certain number of hospitals. These hospitals were excluded from the modeling but represented only 0.4% of the observations in the database. Another limitation of our study is that we could not control for surgeons and hospitals quality. This information is not available in the PMSI database, while these factors could have a significant impact on mortality. This suggests the need for more solid studies, which could consider the effect of the surgeons or the quality of the hospital. The strengths of our study lie in the fact that we used a national administrative billing database, which allowed us to have exhaustive data for the whole of France. The PMSI data also allowed us to follow patients throughout their stays, so we had no missing data at the patient level. The size of our sample was large, and we had enough events, which allowed us to have more accurate estimators. Different models tested in sensitivity analyses gave us results that were consistent with the mains model, attesting to its robustness.

## Conclusions

The results of our studies show that increasing hospital competition independently decreases the 30-day mortality rate after colorectal cancer surgery. Hospital caseload, patients' characteristics and age also impact the post-operative mortality.

## Supporting information

**S1 Appendix. Charlson comorbity score calculation.**
(DOCX)

**S2 Appendix. The codes of common classification of medical acts: 11th version.**
(DOCX)

**S3 Appendix. The codes of international classification of disease: 10<sup>th</sup> version.**
(DOCX)

## Author Contributions

**Conceptualization:** Seydou Goro, Alexandre Challine, Salomé Epaud, Andrea Lazzati.

**Data curation:** Andrea Lazzati.

**Formal analysis:** Seydou Goro, Alexandre Challine, Salomé Epaud.

**Investigation:** Seydou Goro.

**Methodology:** Seydou Goro, Alexandre Challine, Andrea Lazzati.

**Software:** Seydou Goro, Alexandre Challine.

**Supervision:** Andrea Lazzati.

**Validation:** Seydou Goro, Alexandre Challine, Jérémie H. Lefèvre, Andrea Lazzati.

**Visualization:** Seydou Goro, Alexandre Challine, Jérémie H. Lefèvre, Salomé Epaud.

**Writing – original draft:** Seydou Goro, Alexandre Challine, Andrea Lazzati.

**Writing – review & editing:** Seydou Goro, Alexandre Challine, Jérémie H. Lefèvre, Andrea Lazzati.

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
