## [Decision Letter · Decision Letter 0]

23 May 2023

PONE-D-23-0143IMPACT OF INTERHOSPITAL COMPETITION ON MORTALITY OF PATIENTS OPERATED ON FOR COLORECTAL CANCER FACED TO HOSPITAL VOLUME AND RURALITY: CROSS-SECTIONNAL STUDYPLOS ONE

Dear Dr. Challine,

Thank you for submitting your manuscript to PLOS ONE. After careful consideration, we feel that it has merit but does not fully meet PLOS ONE’s publication criteria as it currently stands. Therefore, we invite you to submit a revised version of the manuscript that addresses the points raised during the review process.

The manuscript has been evaluated by two reviewers, and their comments are available below.

The reviewers have raised a number of concerns. Reviewer 1 notes some potential limitations of your study, which you have already noted in your manuscript. Reviewer 2 requests clarification on several points. Could you please carefully revise the manuscript to address all comments raised?

We look forward to receiving your revised manuscript.

Kind regards,

Steve Zimmerman, PhD

Associate Editor, PLOS ONE

Journal Requirements:

2. Thank you for submitting the above manuscript to PLOS ONE. During our internal evaluation of the manuscript, we found significant text overlap between your submission and previous work in the Methods.

Please revise the manuscript to rephrase the duplicated text, cite your sources, and provide details as to how the current manuscript advances on previous work. Please note that further consideration is dependent on the submission of a manuscript that addresses these concerns about the overlap in text with published work.

We will carefully review your manuscript upon resubmission and further consideration of the manuscript is dependent on the text overlap being addressed in full. Please ensure that your revision is thorough as failure to address the concerns to our satisfaction may result in your submission not being considered further.

3. You indicated that ethical approval was not necessary for your study. We understand that the framework for ethical oversight requirements for studies of this type may differ depending on the setting and we would appreciate some further clarification regarding your research. Could you please provide further details on why your study is exempt from the need for approval and confirmation from your institutional review board or research ethics committee (e.g., in the form of a letter or email correspondence) that ethics review was not necessary for this study? Please include a copy of the correspondence as an "Other" file.

6. We note that Figures 1 and 2 in your submission contain map images which may be copyrighted. All PLOS content is published under the Creative Commons Attribution License (CC BY 4.0), which means that the manuscript, images, and Supporting Information files will be freely available online, and any third party is permitted to access, download, copy, distribute, and use these materials in any way, even commercially, with proper attribution. For these reasons, we cannot publish previously copyrighted maps or satellite images created using proprietary data, such as Google software (Google Maps, Street View, and Earth). For more information, see our copyright guidelines: http://journals.plos.org/plosone/s/licenses-and-copyright.

a. You may seek permission from the original copyright holder of Figures 1 and 2 to publish the content specifically under the CC BY 4.0 license.  

Reviewers' comments:

Reviewer's Responses to Questions

**Comments to the Author**

1. Is the manuscript technically sound, and do the data support the conclusions?

Reviewer #1: Yes

Reviewer #2: Yes

2. Has the statistical analysis been performed appropriately and rigorously? 

Reviewer #1: Yes

Reviewer #2: Yes

3. Have the authors made all data underlying the findings in their manuscript fully available?

Reviewer #1: Yes

Reviewer #2: No

4. Is the manuscript presented in an intelligible fashion and written in standard English?

Reviewer #1: Yes

Reviewer #2: No

5. Review Comments to the Author

Reviewer #1: I congratulate the authors for the work made on a large number of patients.

The very accurate results are interesting, as they show the correlation between mortality rate after colon-rectal cancer and the increasing hospital competition, independently of hospital volume and rurality.

On the other hand, the absence of control for surgeons and hospital quality, limits the study, as these two parameters could influence the mortality.

The study is well written in English.

I do not have any further recommendation.

Reviewer #2: There are some typos and grammar errors, i.e., in the title says: cross-sectionnal study instead of “a cross-sectional study”. In the discussion section also says:

. These findings are consistent with a study from previous studies.

. The activity per year per center in non-competitive hospital was higher than in highly competitive center and the surgical activity of the center impact 30-day mortality as previously describe by several authors.

These sentences sound a little weird. I suggest editing help from someone with full professional proficiency in English.

Reference style should be reviewed. In some cases, references are written after the dot, instead of before, i.e,

From 2004 to 2008, pro-competition reforms were gradually introduced.(8)

However, other studies show that competition increases mortality.(13, 14)

Data are not available because there are some restrictions on publicly sharing data. Those are specified (Data were collected from a National Health Database)

The article is interesting and methodologically correct. Statistical analysis also seems the most appropriate for the variables they handled and the objectives of the study. The use of the national database to obtain data from both private and public hospitals allows the sample to be large in order to draw conclusions.

I have some doubts relating to the conclusions. It is indicated that increasing hospital competition decreases the 30-day mortality rate after colorectal cancer surgery independently of the hospital volume and the rurality of the hospital. However, they previously indicate that mortality was higher in hospitals with higher volumes. Besides, the author refers to Amrani's study, which also found an association with hospital volume.

The table also indicates statistical significance of the volume of the hospital:

Hospital’s caseload/year

OR [CI] p.value OR [CI] p.value

>100 0.74 (0.63, 0.86) <0.001 0.69 (0.59, 0.80) <0.001

Instead, hospital status does seem to have had no influence.

I am also left with the question of knowing the differences between the 2 periods studied (before and after the reform) by the Gobillon study that they refer to.

6. PLOS authors have the option to publish the peer review history of their article (what does this mean?). If published, this will include your full peer review and any attached files.

Reviewer #1: No

Reviewer #2: No

---

## [Author Response · Author response to Decision Letter 0]

7 Aug 2023

Response to reviewer

Reviewer #1

I congratulate the authors for the work made on a large number of patients.

The very accurate results are interesting, as they show the correlation between mortality rate after colon-rectal cancer and the increasing hospital competition, independently of hospital volume and rurality.

We thank reviewer 1 for this kind comment.

On the other hand, the absence of control for surgeons and hospital quality, limits the study, as these two parameters could influence the mortality.

We agree with reviewer 1. It could be very interesting to adjust the analysis on surgeon activity volume and control quality of surgeon, and as well add some marker on hospital quality. The database does not contain these datas, we cannot access to the surgeon volume activity nor the rate of completion of text book outcome or the rate of guidelines application for example as we discuss in the limits in the study.

The study is well written in English.

I do not have any further recommendation.

We thanks reviewer 1 for this comment.

Reviewer #2: 

There are some typos and grammar errors, i.e., in the title says: cross-sectionnal study instead of “a cross-sectional study”. In the discussion section also says:

. These findings are consistent with a study from previous studies.

. The activity per year per center in non-competitive hospital was higher than in highly competitive center and the surgical activity of the center impact 30-day mortality as previously describe by several authors.

These sentences sound a little weird. I suggest editing help from someone with full professional proficiency in English.

We thanks the reviewer 2 for these helpful comments, we modify the text according the reviewer comments.

The title :

IMPACT OF INTERHOSPITAL COMPETITION ON MORTALITY OF PATIENTS OPERATED ON FOR COLORECTAL CANCER FACED TO HOSPITAL VOLUME AND RURALITY: A CROSS-SECTIONAL STUDY

In the discussion :

These findings are consistent with previous studies (28-30).

Indeed, the activity volume in a non-competitive hospital may be higher than in a highly competitive center. As surgical activity is a major factor associated to the post-operative mortality, it is very important to take this into account (28-30).

Reference style should be reviewed. In some cases, references are written after the dot, instead of before, i.e,

From 2004 to 2008, pro-competition reforms were gradually introduced.(8)

However, other studies show that competition increases mortality.(13, 14)

We thank the reviewer 2, we have modified the references style according to the comment.

Data are not available because there are some restrictions on publicly sharing data. Those are specified (Data were collected from a National Health Database)

This is true, we can only provide data in acceptable demand according to the ethical French from the CNIL (Commission nationale de l'informatique et des libertés : national commitee of informatic law and liberties)

The article is interesting and methodologically correct. Statistical analysis also seems the most appropriate for the variables they handled and the objectives of the study. The use of the national database to obtain data from both private and public hospitals allows the sample to be large in order to draw conclusions.

We thank the reviewer 2 for this comment.

I have some doubts relating to the conclusions. It is indicated that increasing hospital competition decreases the 30-day mortality rate after colorectal cancer surgery independently of the hospital volume and the rurality of the hospital. However, they previously indicate that mortality was higher in hospitals with higher volumes. Besides, the author refers to Amrani's study, which also found an association with hospital volume.

The table also indicates statistical significance of the volume of the hospital:

Hospital’s caseload/year

OR [CI] p.value OR [CI] p.value

> 100.74 (0.63, 0.86) <0.001 0.69 (0.59, 0.80) <0.001

Instead, hospital status does seem to have had no influence.

We agree with the reviewer comment, hospital volume is major factor of post operative death. 

We modify the discussion of this section. We corrected the impact of volume activity on mortality: high volume activity is correlated to lower mortality as your tables show it :

We also found a significant effect of hospital caseload, patients’ characteristics and age on mortality. These findings are consistent with previous studies, Amrani and al. found an association between lower 90-day post-operative mortality and higher volume activity.

We state that competition is independent factor associated to the mortality, regarding hospital volume and rurality, because after including all these variables in the regression model, there is still a significant association between competition and mortality. We agree, hospital volume had a very significant impact on postoperative mortality. We modify the conclusion about it :

The results of our studies show that increasing hospital competition independently decreases the 30-day mortality rate after colorectal cancer surgery. Hospital caseload, patients’ characteristics and age also impact the post-operative mortality.

I am also left with the question of knowing the differences between the 2 periods studied (before and after the reform) by the Gobillon study that they refer to.

After re-reading this sentence, it adds nothing to the discussion : we decide to remove “Another aspect of Gobillon's study is that it compares the impact of competition on mortality between two periods, the period before the pro-competitive reform and the period after this reform.” from the discussion.

---

## [Decision Letter · Decision Letter 1]

4 Sep 2023

IMPACT OF INTERHOSPITAL COMPETITION ON MORTALITY OF PATIENTS OPERATED ON FOR COLORECTAL CANCER FACED TO HOSPITAL VOLUME AND RURALITY: A CROSS-SECTIONAL STUDY

PONE-D-23-01432R1

Dear Dr. Alexandre Challine,

We’re pleased to inform you that your manuscript has been judged scientifically suitable for publication and will be formally accepted for publication once it meets all outstanding technical requirements.

Kind regards,

Wen-Wei Sung, M.D., Ph.D.

Academic Editor

PLOS ONE

Reviewers' comments:

Reviewer's Responses to Questions

**Comments to the Author**

1. If the authors have adequately addressed your comments raised in a previous round of review and you feel that this manuscript is now acceptable for publication, you may indicate that here to bypass the “Comments to the Author” section, enter your conflict of interest statement in the “Confidential to Editor” section, and submit your "Accept" recommendation.

Reviewer #1: All comments have been addressed

Reviewer #2: All comments have been addressed

2. Is the manuscript technically sound, and do the data support the conclusions?

Reviewer #1: Yes

Reviewer #2: (No Response)

3. Has the statistical analysis been performed appropriately and rigorously? 

Reviewer #1: Yes

Reviewer #2: (No Response)

4. Have the authors made all data underlying the findings in their manuscript fully available?

Reviewer #1: Yes

Reviewer #2: (No Response)

5. Is the manuscript presented in an intelligible fashion and written in standard English?

Reviewer #1: Yes

Reviewer #2: (No Response)

6. Review Comments to the Author

Reviewer #1: the manuscript is well written and correctly conducted.The authors collected a large number of patients that identified the real influence of hospital competition on mortality rate after colorectal cancer

Reviewer #2: The authors have adequately addressed my comments from the previous review round, and I believe this manuscript is now suitable for publication. I would like to point out that regarding the comment I made, "I am also left with the question of knowing the differences between the 2 periods studied (before and after the reform) by the Gobillon study that they refer to", to which they replied, "After re-reading this sentence, it adds nothing to the discussion, we decide to remove 'Another aspect of Gobillon's study is that it compares the impact of competition on mortality between two periods, the period before the pro-competitive reform and the period after this reform.' from the discussion." However, this paragraph remains in the revised article; it has not been removed.

7. PLOS authors have the option to publish the peer review history of their article (what does this mean?). If published, this will include your full peer review and any attached files.

Reviewer #1: **Yes: **Nora Di Tomasso

Reviewer #2: No

---

## [Editor Report · Acceptance letter]

11 Sep 2023

PONE-D-23-01432R1 

IMPACT OF INTERHOSPITAL COMPETITION ON MORTALITY OF PATIENTS OPERATED ON FOR COLORECTAL CANCER FACED TO HOSPITAL VOLUME AND RURALITY: A CROSS-SECTIONAL STUDY 

Dear Dr. Challine:

I'm pleased to inform you that your manuscript has been deemed suitable for publication in PLOS ONE. Congratulations! Your manuscript is now with our production department. 

Kind regards, 

on behalf of

Dr. Wen-Wei Sung 

Academic Editor

PLOS ONE